# In Vitro Antifungal Activity of Three Synthetic Peptides against *Candida auris* and Other *Candida* Species of Medical Importance

**DOI:** 10.3390/antibiotics12081234

**Published:** 2023-07-26

**Authors:** Richar Torres, Adriana Barreto-Santamaría, Gabriela Arévalo-Pinzón, Carolina Firacative, Beatriz L. Gómez, Patricia Escandón, Manuel Alfonso Patarroyo, Julián E. Muñoz

**Affiliations:** 1Faculty of Health Sciences, Universidad Colegio Mayor de Cundinamarca, Bogotá 110311, Colombia; richar-duvan@hotmail.com; 2Studies in Translational Microbiology and Emerging Diseases (MICROS) Research Group, School of Medicine and Health Sciences, Universidad de Rosario, Bogotá 111221, Colombia; cfiracative@gmail.com (C.F.); beatriz.gomez@urosario.edu.co (B.L.G.); 3Receptor-Ligand Department Fundación Instituto de Inmunología de Colombia (FIDIC), Bogotá 111321, Colombia; adrianasantamaria10@gmail.com (A.B.-S.); gabarpi@gmail.com (G.A.-P.); 4Microbiology Group, Instituto Nacional de Salud, Bogotá 111321, Colombia; pescandon@ins.gov.co; 5Molecular Biology and Immunology Department, Fundación Instituto de Inmunología de Colombia (FIDIC), Bogotá 111321, Colombia; mapatarr.fidic@gmail.com; 6Microbiology Department, Faculty of Medicine, Universidad Nacional de Colombia, Bogotá 111321, Colombia; 7Public Health Research Group, School of Medicine and Health Sciences, Universidad del Rosario, Bogotá 111221, Colombia

**Keywords:** *Candida auris*, *Candida* spp., antimicrobial peptides, antifungal resistance, candidiasis

## Abstract

Candidiasis is an opportunistic infection affecting immunosuppressed and hospitalized patients, with mortality rates approaching 40% in Colombia. The growing pharmacological resistance of *Candida* species and the emergence of multidrug-resistant *Candida auris* are major public health problems. Therefore, different antimicrobial peptides (AMPs) are being investigated as therapeutic alternatives to control candidiasis effectively and safely. This work aimed to evaluate the in vitro antifungal activity of three synthetic AMPs, PNR20, PNR20-1, and 35409, against ATCC reference strains of *Candida albicans*, *Candida glabrata*, *Candida parapsilosis*, *Candida krusei*, and *Candida tropicalis*, and clinical isolates of *C. auris*. Antifungal susceptibility testing, determined by broth microdilution, showed that the AMPs have antifungal activity against planktonic cells of all *Candida* species evaluated. In *C. auris* and *C. albicans*, the peptides had an effect on biofilm formation and cell viability, as determined by the XTT assay and flow cytometry, respectively. Also, morphological alterations in the membrane and at the intracellular level of these species were induced by the peptides, as observed by transmission electron microscopy. In vitro, the AMPs had no cytotoxicity against L929 murine fibroblasts. Our results showed that the evaluated AMPs are potential therapeutic alternatives against the most important *Candida* species in Colombia and the world.

## 1. Introduction

Antimicrobial resistance is a major problem that concerns health and scientific authorities around the world, including Colombia. Despite continuous efforts to develop new therapeutic agents, the resistance profiles of many pathogenic microorganisms are increasingly worsening, leading to a significantly reduced effectiveness of the available antimicrobials. This is why it is essential to search for and develop new alternative antimicrobial agents with different mechanisms of action that allow us to deal with this threat to global public health [1].

Among the invasive fungal infections occurring in Colombia, candidiasis alone accounts for 88% of cases, frequently causing bloodstream infections (BSI) in hospitalized patients and being associated with high mortality rates of about 40% [2]. In addition, effective prevention and treatment of this mycosis are also threatened by the increasing incidences of resistant species, mainly *Candida albicans*, but also non-*albicans* species such as *Candida glabrata*, *Candida parapsilosis*, *Candida krusei*, *Candida tropicalis*, and *Candida auris*. Particularly this last species has gained importance since the infections it causes are associated with high rates of morbidity and mortality, which could be attributed to the fact that *C. auris* presents high rates of resistance to conventional antifungals, which can easily spread from patient to patient in hospital settings and is difficult to identify using traditional laboratory methods due to its biochemical and morphological resemblance to other *Candida* species, such as those of the *Candida haemulonii* complex [1,3].

The resistance profile of *C. auris* together with its degree of pathogenicity, which is similar to that of *C. albicans*, and its ability to persist for prolonged periods in the environment, have contributed to the worldwide spread of this microorganism since it was first described in 2009 [4,5,6]. Thus, *C. auris* has emerged as an important nosocomial pathogen, causing between 5% and 30% of all candidemia cases worldwide, affecting patients mainly in intensive care units (ICUs), and being associated with high crude mortalities ranging between 33% and 100%, which are the result of therapeutic failure due to antifungal resistance [3,7,8].

In Colombia, the National Institute of Health (NIH) together with the Ministry of Health and Social Protection issued the notification “Global emergency alert of invasive infections caused by the multi-resistant yeast *Candida auris*” in September 2016. Later, in July 2017, the document “Broadsheet to strengthen surveillance actions for the emerging multi-resistant yeast *C. auris*” was published [9,10,11]. From then until 2019, the NIH in Colombia received 689 *C. auris* clinical isolates associated with invasive infections, of which about 40% are resistant to amphotericin B and about 60% to fluconazole, two of the most widely used antifungals to control candidiasis in the country and worldwide [12,13].

Additionally, outbreaks of *C. auris* infection have been characterized in at least 20 medical institutions in 16 departments of the country, with an estimated crude mortality of 50% and affecting patients with a wide age range. In addition, the characterization of these outbreaks led to the recovery of the yeast from the hospital environment, corroborating the fact that this multidrug-resistant species has the ability to contaminate the environment close to patients, which is highly important in terms of healthcare-associated infections [14].

When comparing the number of drugs available to treat bacterial infections against those to fight mycoses, it is evident that the latter group is much more limited and less diverse. This is why it is crucial to seek and develop new therapeutic agents against fungi, particularly against multidrug-resistant microorganisms such as *C. auris* [15]. Antimicrobial peptides (AMPs), which have emerged as a therapeutic option against resistant pathogens, represent a promising alternative to combat fungal infections [16]. Among the main advantages of AMPs is their ability to break the cell membrane of microorganisms, which is a target with a low rate of modifications compared to other cell targets. Similarly, activity on the membrane has been associated with faster antimicrobial mechanisms of action and with less development of resistance [17].

Recent research carried out at the Fundación Instituto de Inmunología de Colombia (FIDIC) assessed two synthetic peptides obtained by computational design, PNR20 (1609) and PNR20-1 (29009), finding that these peptides have antibacterial activity against both Gram-negative and Gram-positive bacteria (unpublished results). In addition, the synthetic peptide 35409, which is an analog derived from peptide 20628 obtained from the rifin of *Plasmodium falciparum* (*Pf*Rif protein) [18], was also shown to have antibacterial activity against Gram-negative bacteria, such as *Escherichia coli*, being associated with a membranolytic effect [18]. Overall, the characteristics of the peptides PNR20, PNR20-1, and 35409, such as their high content of helical structures, positive charge, and content of hydrophobic amino acids, fit within the parameters described for AMPs with high action potential on negatively charged membranes, such as fungal membranes. Therefore, this study aimed to establish the antifungal capacity of peptides PNR20, PNR20-1, and 35409 against reference stains of *Candida* species of clinical importance in Colombia and the world, such as *C. albicans*, *C. glabrata*, *C. parapsilosis*, *C. krusei*, and *C. tropicalis*, as well as Colombian clinical isolates of *C. auris*.

## 2. Results

### 2.1. Peptides PNR20, PNR20-1, and 35409 Present Differential Antifungal Activity against Most Candida Species Studied

The antifungal activity of the antimicrobial peptides PNR20, PNR20-1, and 35409 against planktonic cells of reference strains of *C. albicans*, *C. parapsilosis*, *C. glabrata*, *C. krusei,* and *C. tropicalis* and clinical isolates of *C. auris* is shown in Table 1.

In general, the evaluated peptides showed antifungal activity against all *Candida* strains evaluated, inhibiting growth by 50% (IC_50_) or reaching total inhibition. Considering the concentrations of the peptides tested, it was possible to determine that the peptide PNR20 exhibited the broadest spectrum of antifungal activity, as it inhibited the total growth of all five reference strains of *Candida*, with the minimum inhibitory concentration (MIC) ranging from 25 µM to 100 µM. The peptide PNR20-1, which totally inhibited the growth of three reference strains, had the second broadest spectrum, while the peptide 35409, inhibiting two reference strains, was the peptide with the narrowest spectrum of antifungal activity.

Among the species, the most susceptible strain to the activity of the peptides was *C. krusei* ATCC 6558 since the three peptides were able to inhibit 100% of yeast growth, with the lowest MIC value reported in this study, 25 µM, for PNR20 and PNR20-1, and 100 µM for 35409. The total growth of *C. tropicalis* ATCC 750 was also inhibited by the three peptides, with MIC values of 50 µM for PNR20 and 35409 and 100 µM for PNR20-1 (Table 1). *C. albicans* ATCC 10231 growth was totally inhibited by 100 µM of each of the peptides PNR20 and PNR20-1, while the total growth of *C. parapsilosis* ATCC 22019 and *C. glabrata* ATCC 2001 was only achieved by 100 µM of the peptide PNR20.

Against the clinical isolates of *C. auris*, total growth inhibition was not achieved with any of the peptides and concentrations tested. However, all the peptides evaluated were capable of inhibiting 50% of cellular growth of all isolates studied, with peptide PNR20 being the one that showed a broader spectrum of antimicrobial activity. Peptides PNR20-1 and 35409 had antifungal activity (IC_50_) on three and two isolates of *C. auris*, respectively.

### 2.2. Peptides PNR20 and PNR20-1 Inhibit Biofilm Formation in C. albicans and C. auris

The inhibitory activity of peptides PNR20, PNR20-1, and 35409 against metabolic activity in *C. albicans* ATCC 10231 and *C. auris* H0059-13-1421 is shown in Table 2. In *C. albicans*, peptides PNR20 and PNR20-1 inhibited 100% of biofilm formation at 24 h in concentrations that were even lower than the one determined in the inhibition assay with planktonic cells. At 72 h, the same inhibitory behavior of the peptides PNR20 and PNR20-1 was observed with BMIC values between 2 and 4 times higher, respectively, than those described in an incubation time of 24 h.

With *C. auris* H0059-13-1421, the three studied peptides were able to inhibit up to 50% of biofilm formation after both 24 and 72 h of exposure. The inhibitory concentrations presented slight variations between the incubation times at 24 and 72 h, although the inhibition was generally similar.

### 2.3. Peptides PNR20, PNR20-1, and 35409 Affect the Cellular Morphology of C. albicans and C. auris

The evaluated peptides had the ability to alter the cellular structure of *C. albicans* ATCC 10231 when compared to the control (yeasts without treatment). Yeasts treated with peptides PNR20-1 and 35409 showed visible alterations in their cellular structure. In addition, the three peptides clearly disrupted the membrane, causing the cell to lose its regular spherical shape (Figure 1). Additionally, the organelles in the treated cells were observed to be altered, which is a sign of a significant intracellular disturbance.

When *C. auris* H0059-13-1421 was tested, the three peptides evaluated had a slightly different effect (Figure 2). In general, with respect to yeasts without treatment, the treated cells maintained their regular spherical shape, and no clear disturbances in the continuity of the membrane were observed. However, at the intracellular level, it was observed that the organelles were altered, and even with peptide 35409, an accumulation of small vacuoles was observed at its left end.

### 2.4. C. albicans Viability Is Altered after Treatment with Peptides PNR20, PNR20-1, and 35409

The cell viability of *C. albicans* ATCC 10231 yeasts that were treated with the AMPs PNR20, PNR20-1, and 35409 decreased compared to the untreated cells. The most notable reduction in viability was observed in the yeasts treated with the peptide 35409, which presented 58% mortality, followed by the yeasts treated with PNR20-1 with 49% of the dead cells and finally the yeasts treated with PNR20, which showed a mortality of 16%. Figure 3 shows that the cell viability of yeasts is affected when treated with AMPs.

### 2.5. Peptides PNR20, PNR20-1, and 35409 Are Not Cytotoxic toward Murine Fibroblasts

The studied peptides PNR20, PNR20-1, and 35409 did not show cytotoxicity toward L929 cells (murine fibroblasts) treated for 24 h with a concentration of 200 µM for each peptide. Figure 4 shows that the cell viability of L929 cells was not affected by any of the three peptides.

## 3. Discussion

In Colombia, candidiasis remains one of the most prevalent invasive fungal infections. Between 2010 and 2013, 94.5% of the fungal species causing invasive infections in patients in ICUs of 20 tertiary care institutions in 10 Colombian cities were identified as *Candida* species, with *C. albicans* being the most frequent agent (48.3%), followed by *C. tropicalis* (38.6%) and *C. parapsilosis* (28.5%). Additionally, it was estimated that *Candida* species are responsible for about 25% of healthcare-associated infections in the country, mainly due to catheter-associated BSI [19].

Depending on the clinical presentation and the species of *Candida* that is isolated in the laboratory, the treatment differs, with azoles being one of the main options [12]. Within this family of antimycotic drugs, fluconazole is widely used. However, in Colombia, resistance to this triazole has been reported to be around 35% in isolates of *C. auris* [11]. In addition, resistance to voriconazole, another therapeutic option used in the hospital practice, has been reported to be increasing over time in Colombia. Between 2001 and 2007, 16% of clinical isolates of *Candida* were found to be resistant to voriconazole, while between 2010 and 2011, resistance to this azole increased to 26% of isolates [20,21]. On the other hand, *C. auris* presents a resistance of 33% to amphotericin B, another antifungal of choice for the treatment of candidiasis [11].

Under this scenario, AMPs become an interesting alternative to address this progressive problem of drug resistance in *Candida* species. Not only has the antifungal activity of AMPs on free cells been described, but also on the formation of biofilms, an important virulence factor of *Candida* yeasts; in addition, the safety of many of these AMPs in mammalian cells has been shown in in vitro and in vivo cytotoxicity assays [22,23].

Considering that the antibacterial capacity of three synthetic peptides, PNR20, PNR20-1, and 35409, has been reported on Gram-negative and Gram-positive bacteria (unpublished results) [18], our study aimed to establish whether the antimicrobial effect could possibly extend to yeasts of public health importance due to the similarity of electrical charge between bacterial and fungal membranes [24]. As such, these peptides were tested in the most frequent *Candida* species associated with candidiasis in Colombia and worldwide.

With *C. albicans* ATCC 10231, it was observed that two of the three peptides (PNR20 and PNR20-1) managed to inhibit 100% of cell growth at a concentration of 100 µM in both molecules. This value is not far from those reported previously [25], since inhibitory concentrations between 79 and 105 µM (150 to 200 µg/mL) were obtained for the indolicin peptide in different strains of *Candida* species.

For *C. parapsilosis* ATCC 22019, the only peptide that was able to completely inhibit yeast growth was PNR20, at a concentration of 100 µM. This MIC value is higher than that reported before with other peptides such as Jelleine-I, which prevented the total growth of this yeast at a concentration of 64 µM [26]. However, it is important to consider that modifications or structural differences of the antimicrobial peptides may or may not improve the antimicrobial response or make them more or less cytotoxic.

The MIC value reported in this study with peptide PNR20 was 100 µM for *C. glabrata* ATCC 2001. Other studies have described complete growth inhibition of this species at concentrations between 32 µM and 50 µM [26,27]. However, different studies have also described that *C. glabrata* is the most resistant *Candida* species to the activity of AMPs, the reason for this phenomenon being unknown [28,29]. Therefore, any antifungal activity with antimicrobial peptides such as that described with PNR20 would be of interest to study further to treat the naturally resistant *C. glabrata*.

Contrary to the previous case, *C. krusei* ATCC 6558 was the most susceptible species to the three antimicrobial peptides evaluated in this study. All peptides inhibited yeast growth by 100%, with PNR20 and PNR20-1 presenting better activity with a MIC value of 25 µM. This result is superior to the activity of Jelleine-I, whose MIC for this species was 32 µM. Other AMPs analyzed with *C. krusei* have presented MICs similar to those observed in this study, such as the antifungal capacity of the archaeal cryptic peptide VLL-28 [26,28]. *C. krusei* is naturally resistant to fluconazole and usually acquires other resistance mechanisms against other types of antifungals [30]. Despite this, several reports have also described that this species of *Candida* is usually more susceptible to the antifungal activity of AMPs [28]. It is plausible that the reason for this phenomenon lies in its structural and metabolic characteristics compared to other *Candida* species [31], but there are still no studies that validate or refute this assumption.

For *C. tropicalis* ATCC 750, the three peptides were able to inhibit 100% of the growth in concentrations of 50 µM (PNR20 and 35409) and 100 µM (PNR20-1), this being the second most susceptible species to the activity of the peptides studied after *C. krusei* ATCC 6558. The results described in this study are similar to those reported for cyclic L-temporin, whose MIC was also 50 µM [27]. Compared to the MIC described with another peptide such as indolicidin, which is 79 µM (150 µg/mL), two of the peptides included in this investigation have therefore better activity [25].

*C. auris* is the species of *Candida* that has been attracting the most attention worldwide in the last decade, due to its appearance as a human pathogen that is naturally resistant or multi-resistant to traditional antifungals [8]. In our study, and together with *C. glabrata* ATCC 2001, *C. auris* was the most resistant species to the activity of the peptides evaluated, since the highest concentration evaluated in of all of them did not inhibit its growth completely, but only managed to inhibit up to 50% of yeast growth.

However, the IC_50_ result obtained in this study for the fluconazole-susceptible isolate H0059-13-1421 is better than that presented in other studies. For example, crotamine peptide inhibited 50% of *C. auris* CBS 1091 growth at a concentration of 40 µM [32], while peptides PNR20-1 and 35409 had IC_50_ concentrations of 6.25 µM and 12.5 µM, respectively. Even the IC_50_ concentration of the PNR20 peptide is remarkable since the activity of the peptide starts at 25 µM.

Regarding the *C. auris* clinical isolates that are resistant to fluconazole, our results showed a behavior similar to that observed in the fluconazole-susceptible strain of *C. auris*, that is, all the studied peptides inhibited up to 50% of yeast growth. However, peptide PNR20 was the only one that inhibited growth in the four resistant strains, even maintaining the same IC_50_ in *C. auris* H0059-13-2220 (25 µM). For the rest of the strains and peptides, the 50% inhibitory concentrations were between 50 µM and 100 µM. Compared with the snake venom-derived peptide, crotamine, our results against fluconazole-resistant clinical isolates of *C. auris* are of interest. With the mentioned peptide, the IC_50_ values were between 80 µM and 160 µM, and thus significantly higher than those reported in our study for peptides PNR20, PNR20-1, and 35409 [32].

Apart from the characteristic of being fluconazole-resistant, the four isolates of *C. auris* included in this study also presented reduced susceptibility to the peptides PNR20-1 and 35409. Interestingly, some species of *Candida* have been reported to secrete certain proteases, such as the Mp65 mannoprotein or the Sap6 aspartic protease, as a defense mechanism against the action of AMPs [33]. This does not imply, however, that it is the only defense mechanism of *Candida* species against AMPs or that this is the explanation for the innocuousness of peptides PNR20-1 and 35409; perhaps the *C. auris* isolates studied here might also be developing mechanisms of defense from the synthetic peptides.

On the other hand, it is striking that the peptide PNR20 is the only one with activity in all strains of *C. auris* that are resistant to fluconazole. Peptide PNR20 differs by having a lysine, a polar amino acid, in position 10, compared to peptide PNR20-1, which presents instead a nonpolar residue, alanine. This is interesting considering the fact that polar amino acids have been used to enhance the activity of antimicrobial peptides [34]. However, 60% of the amino acid structure sequence of 35409 is made up of polar amino acids and is cationic, like PNR20 and PNR20-1 [18]. This implies that there are more structural and chemical factors that influence the activity of an antimicrobial peptide.

Peptides PNR20 and PNR20-1 were able to inhibit biofilm formation in *C. albicans* ATCC 10231 with concentrations of 12.5 µM and 25 µM, respectively, in 24 h, even extending their effect to 72 h with concentrations of 25 µM for PNR20 and 100 µM for peptide PNR20-1. These results are important considering that biofilm formation is an important virulence factor in *Candida* species, particularly *C. albicans* [35]. Our results show better anti-biofilm activity compared to other antimicrobial peptides such as ToAP2, whose inhibition of biofilm formation in *C. albicans* was reached at 25 µM, or analogous peptides derived from bee venom that inhibited up to 90% of biofilm formation at a concentration of 28.6 µM for LL-III/43 and 62.3 µM for VIII [29,36].

Regarding *C. auris*, previous studies have shown that this species, compared to *C. albicans*, formed significantly reduced production of biofilms [37,38]. In our study, the anti-biofilm activity of antimicrobial peptides in *C. auris* H0059-13-1421 was assessed. Peptides PNR20 and PNR20-1 inhibited biofilm formation by up to 50% in 24 h at concentrations up to 25 µM, while peptide 35409 had an anti-biofilm effect at 12.5 µM. Our results also showed that the peptides maintained their inhibitory activity at 72 h, except for 35409, which inhibited 50% of biofilm formation at a concentration of 25 µM. Other peptides, such as cyclic L-temporin analogues, managed to inhibit 90% of biofilm formation in *C. auris* with the same concentration of 25 µM, indicating that the antifungal potential of the three peptides selected in this study is promising [27].

Transmission electron microscopy images showed a significant disturbance in the ultrastructure or morphology of *C. albicans* ATCC 10231 and *C. auris* H0059-13-1421 cells after 24 h of treatment with the three antimicrobial peptides. In both species, significant alterations were observed inside the cell, being more severe in *C. albicans*, which is logical considering that the peptides had greater inhibitory activity for this species compared to *C. auris*. The morphological and intracellular disturbances observed in yeasts treated with antimicrobial peptides are generally related to alterations in the cell membrane and wall, the condensation of genetic material, and the production of cell inclusions, among others [39]. For example, with the ToAP2 peptide, significant alterations were evident in the membrane of the treated yeasts in such a way that the typical oval shape of these cells was lost, while with the LL-III/43 and VIII peptides, there were clear signs of intracellular alterations [29,36].

Other studies have shown that AMPs can exert antimicrobial effects without affecting membrane integrity. A clear example of these alternative mechanisms is that of the Jelleine-I peptide, whose inhibitory activity is the result of the production of reactive oxygen species inside the cell, or the KP peptide, which also stimulates the formation of these compounds, although with an additional disruptive action on mitochondria [26,34]. More recent investigations have also shown that the inhibitory effect could be exerted directly on the DNA of *Candida* spp. The MAF-1A peptide had the ability to decrease the expression of the *ERG5*, *ERG6*, and *ERG11* genes, which are associated with ergosterol biosynthesis and are vital for the formation of the cell membrane [40].

The analysis of cell viability with propidium iodide showed a decrease in the viability of the yeasts treated with the AMPs PNR20, PNR20-1, and 35409. These analyses, utilizing flow cytometry, are important to be able to quantify the viability of different microorganisms, such as *C. albicans,* exposed to antimicrobial substances or molecules, as was observed with the use of metal complex molecules, such as dtc1 and dtc3, which have low toxicity and potent antifungal activity against *C. albicans* [41].

The murine fibroblast cell line L929 treated with the peptides PNR20, PNR20-1, and 35409 did not present changes in its regular growth, demonstrating the innocuousness of the peptides for these cells. Different studies have described no cytotoxicity with other cell lines, such as D-Cateslitin on human gingival cells or the Kw4 peptide on human HaCaT keratinocytes treated with other AMPs [33,42].

To conclude, the results presented in this work show that the peptides PNR20, PNR20-1, and 35409 have in vitro antifungal activity against clinically important *Candida* species, including the multidrug-resistant *C. auris*. As such, these peptides could eventually be considered potential therapeutic alternatives for the control of *Candida* infections, not only because of their inhibitory potential on planktonic cells of diverse species of *Candida*, but also because of the safety of the peptides, since they had no cytotoxic effects on the reference mammalian cell line L929. Interestingly, this study highlights the major antifungal effect of peptide PNR20, compared to the other peptides evaluated, even in species such as *C. auris.* It is worth noting that the majority of the results described here were obtained with ATCC reference strains. We are currently carrying out experiments aimed at assessing whether similar results can be obtained when testing these AMPs in isolates recovered from clinical samples. In addition, possible synergic interactions between the assessed AMPs and commonly used antifungals, such as fluconazole, will be explored, particularly with isolates that showed reduced susceptibility to the AMPs alone.

## 4. Materials and Methods

### 4.1. Synthesis and Purification of Antimicrobial Peptides

The peptides PNR20 (1609), PNR20-1 (29009) (unpublished results), and 35409 (RYRRKKKMKKALQYIKLLKE) used in this study are 20-mer peptides [18]. Peptide PNR20 was obtained by random selection of 10 polar and 10 nonpolar amino acids. Peptide PNR20-1, analog peptide of PNR20, has a lysine–alanine substitution at position 10 (unpublished results). Peptide 35409 is an analog derived from *Pf*Rif protein [18]. After design, peptides were synthesized by Peptide 2.0 (Chantilly, VA, USA). HPLC and MS analyses performed by the manufacturer, which showed that the synthetic peptides were 95% pure.

### 4.2. Microorganisms and Cells

The reference strains of *C. albicans* ATCC 10231, *C. parapsilosis* ATCC 22019, *C. krusei* ATCC 6558, *C. tropicalis* ATCC 750, and *C. glabrata* ATCC 2001, from the American Type Culture Collection (ATCC) (Manassas, VI, USA), were used in this study. In addition, five clinical isolates of *C. auris*, including one that was susceptible to fluconazole and amphotericin B (isolate H0059-13-1421) and four that were resistant to both fluconazole and amphotericin B (isolates H0059-13-2220, H0059-13-2251, H0059-13-2276, and H0059-13-2265), were also tested. Although CLSI does not provide break points for *C. auris*, isolates with minimum inhibitory concentration (MIC) ≥ 2 μg/mL and ≥32 μg/mL for amphotericin B and fluconazole, respectively, were considered resistant [43]. Clinical isolates were recovered from blood cultures and urine. All strains were stored in 10% glycerol at −80 °C. Three days before each experiment, the strains were cultured separately on Sabouraud dextrose agar (Becton, Dickinson, New Jersey, NJ, USA) and incubated at 37 °C for 24 to 48 h in order to recover exponentially growing yeasts.

The murine fibroblast cell line L929 was obtained from the ATCC (Manassas, VI, USA) and kept in liquid nitrogen at −180 °C. At the time of the experiments, the cells were thawed and placed in Dulbecco’s Modified Eagle Medium (DMEM) culture medium (Gibco, Waltham, MA, USA) with 10% fetal bovine serum (Gibco, Waltham, MA, USA) and 1% penicillin–streptomycin (10,000 U/mL) (Gibco, Waltham, MA, USA).

### 4.3. Susceptibility Assay of Planktonic Cells of Candida

Susceptibility testing of planktonic cells of each *Candida* species against each peptide was determined using broth microdilution, as described in the document M27-S4 of the Clinical and Laboratory Standards Institute (CLSI) [44]. Briefly, yeasts were suspended in 10 mL of RPMI 1640 medium (BiowHITTAKER^®^, Lonza, Belgium) supplemented with 3-(n-morpholino) propanesulfonic acid (MOPs) (Sigma-Aldrich, St. Louis, MO, USA), and the inoculum was adjusted to the turbidity of a 0.5 McFarland standard, which represents a concentration of 1–5 × 10^3^ yeast cells per milliliter. Peptides PNR20, PNR20-1, and 35409 were adjusted separately to concentrations of 100 µM, 50 µM, 25 µM, 12.5 µM, 6.25 µM, 3.12 µM, 1.56 µM, 0.78 µM, 0.39 µM, and 0.19 µM in RPMI-MOPs medium. Subsequently, 100 µL of each concentration was deposited, per triplicate, in sterile 96-well round-bottom polystyrene microtiter plates (Corning Incorporated, New York, NY, USA) and mixed with 100 µL of each inoculum. Fluconazole (FCZ) (Pfizer, New York, NY, USA), at concentrations ranging by 2-fold dilutions from 0.5 µg/mL to 128 μg/mL, was used as a control antifungal since the inhibitory concentrations of this drug against the reference strains are known and this antifungal is frequently used in the treatment of candidiasis (7,18). Per plate, 100 µL of the inoculums of each of the studied isolates plus 100 µL of RPMI-MOPs, without peptides, were incubated as growth controls. Wells with media but without peptides and inoculum, were used as sterility controls. After 48 h of incubation at 37 °C, optical density (OD) at 405 nm was measured in each well of the plates using a Multiskan FC spectrophotometer (Thermo Fisher Scientific Inc., Waltham, MA, USA). The minimum inhibitory concentration (MIC), defined as the lowest concentration of the peptides capable of inhibiting the total growth of the different *Candida* species evaluated in this study, was determined per peptide and per species. In some cases, the concentration of the peptides that inhibited the growth of cells by 50% (IC_50_) and by 90% (IC_90_) was also established.

### 4.4. Inhibition of Biofilm Formation Assay

The capacity of the peptides to inhibit biofilm formation was evaluated according to a previously described protocol [45]. For this experiment, the reference strain of *C. albicans* ATCC 10231 and the FCZ-susceptible isolate of *C. auris* H0059-13-1421 were used, considering their good capacity to form biofilm. Initially, each isolate was growth in Sabouraud dextrose broth (Becton, Dickinson and Company; Sparks, NV, USA), for 24 h at 37 °C in shaker at 100 rpm. Later, an inoculum with a concentration of 2 × 10^6^ cells/mL was prepared in RPMI-MOPs medium, and 50 μL of the inoculum was added separately to each well of a 96-well flat-bottom microdilution plate (Costar, Corning Incorporated, New York, NY, USA). Subsequently, 50 μL of different concentrations (0.19 µM to 100 µM) of the antimicrobial peptides PNR20, PNR20-1, and 35409 was added to the plates, which were then incubated for 24 h and 72 h at 37 °C. After the incubation process, the supernatant was removed and the biofilm was quantified by the 2,3-bis-(2-methoxy-4-nitro-5-sulfophenyl)-2H-tetrazolium-5-carboxanilide (XTT) reduction assay described previously to determine the biofilm minimum inhibitory concentration (BMIC) by determining OD values at 492 nm [46].

### 4.5. Evaluation of Cell Morphology by Transmission Electron Microscopy

Cell morphology of the reference strain of *C. albicans* ATCC 10231 and the FCZ-susceptible isolate of *C. auris* H0059-13-1421 was evaluated by transmission electron microscopy (TEM) after treatment with peptides PNR20, PNR20-1, and 35409. A suspension of each isolate was prepared in distilled water and adjusted to a concentration of 1–5 × 10^6^ cell/mL. From this inoculum, 500 μL was added to a 1.5 mL Eppendorf tube, mixed with 50 μL of each antimicrobial peptide (100 μM), and incubated at 37 °C for 24 h. Subsequently, cells were harvested and fixed in 2.5% glutaraldehyde (Sigma-Aldrich, St. Louis, MO, USA) and 4% paraformaldehyde (Sigma-Aldrich, St. Louis, MO, USA). Cells were washed with the same solution and fixed with 1% osmium tetroxide (Sigma-Aldrich, St. Louis, MO, USA), dehydrated by acetone gradient, and resin-embedded. Finally, the cells were stained with uranyl acetate (Thermo Fisher Scientific Inc., Waltham, MA, USA), examined in a transmission electron microscope (JEOL JEM-1400 Plus, Tokyo, TYO, Japan), and photographed with a Gatan camera (Pleasanton, CA, USA) in the pathology laboratory of the Santa Fe Foundation, in Bogotá. Yeasts without treatment with antimicrobial peptides were used as control.

### 4.6. Evaluation of Viability by Flow Cytometry

*C. albicans* ATCC 10231 (1 × 10^6^ CFU/mL) was incubated with 100 μM of the peptides PNR20, PNR20-1, and 35409 (500 μL final volume) for 24 h at 37 °C. The peptide–yeast mixture was then incubated with 10 μg/mL of propidium iodide (PI) for 30 min in the dark [18]. Fluorescence was read by FACSCanto II (Beckton Dickinson), and the flow cytometer (4-2-2 configuration) FlowJo™ Software version 10.9 was used to analyze the data. Dead yeast cells, obtained by heat treatment (5 min at 100 °C and 3 h at 70 °C), and yeasts without any treatment were used for establishing cut-off points between yeast with permeabilized membranes and living ones.

### 4.7. In Vitro Cytotoxicity Assay with the Murine Cell Line L929

The cytotoxicity of peptides PNR20, PNR20-1, and 35409 was evaluated in the murine fibroblast cell line L929 model cultured in DMEM medium and enriched with 1 g/L of D-glucose, 200 mM of L-glutamine, and 110 mg/L of sodium pyruvate (Gibco, Waltham, MA, USA). A total of 100 μL of the cell suspension was seeded in 96-well flat-bottom plates (3 × 10^5^ cells/well) and incubated at 37 °C in 5% CO_2_ for 24 h. Subsequently, 100 μL of different concentrations of the peptides 35409, PNR20, and PNR20-1 (12.5 μM to 200 μM) were added separately to the wells and incubated at 37 °C in 5% CO_2_ for 24 h. The toxicity of the peptides was established by determining the cell viability of L929, which was measured using the Thiazolyl Blue Tetrazolium Bromide (MTT) method (Sigma, St. Louis, MO, USA), according to an established protocol [47]. The controls used for this assay were growth control (cells without treatment) and death control (cells treated with 100% dimethyl sulfoxide (DMSO) (Sigma-Aldrich, St. Louis, MO, USA).

### 4.8. Statistical Analysis

For biofilm and cytotoxicity assays, statistical were performed using GraphPad Prism version 9.0 (GraphPad Software, San Diego, CA, USA). Statistical comparisons were carried out using analysis of variance (one-way ANOVA), followed by a Tukey–Kramer post hoc test. According to the methods used, OD values and percentage of cell viability were compared. *p* values of < 0.05 indicated statistical significance.

## Figures and Tables

**Figure 1 antibiotics-12-01234-f001:**
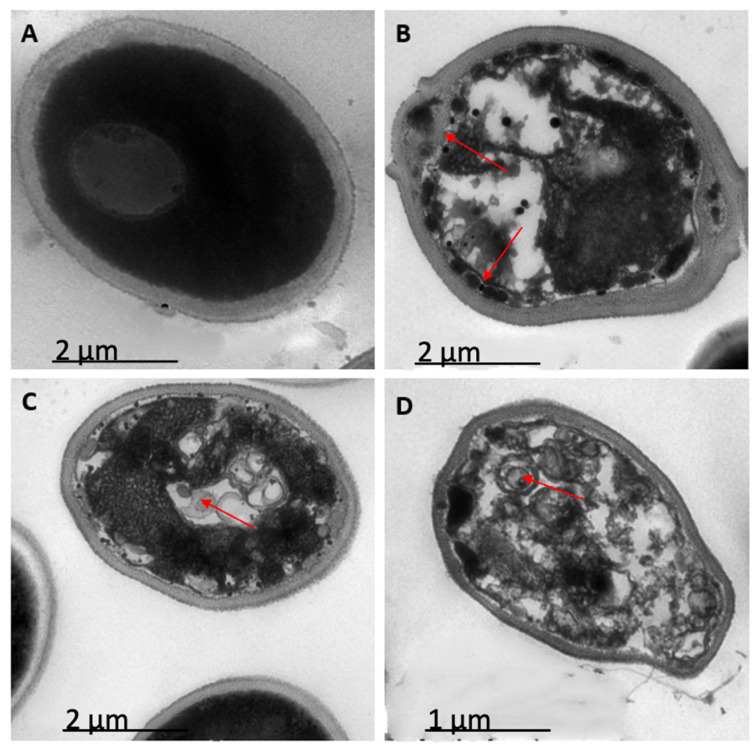
Ultrastructure of yeast cells of *Candida albicans* ATCC 10231, untreated (**A**) and treated, with 100 µM of each of the peptides PNR20 (**B**), PNR20-1 (**C**), and 35409 (**D**). Red arrows show morphological alterations. Microphotographs were taken with Gatan camera in a transmission electron microscope (TEM).

**Figure 2 antibiotics-12-01234-f002:**
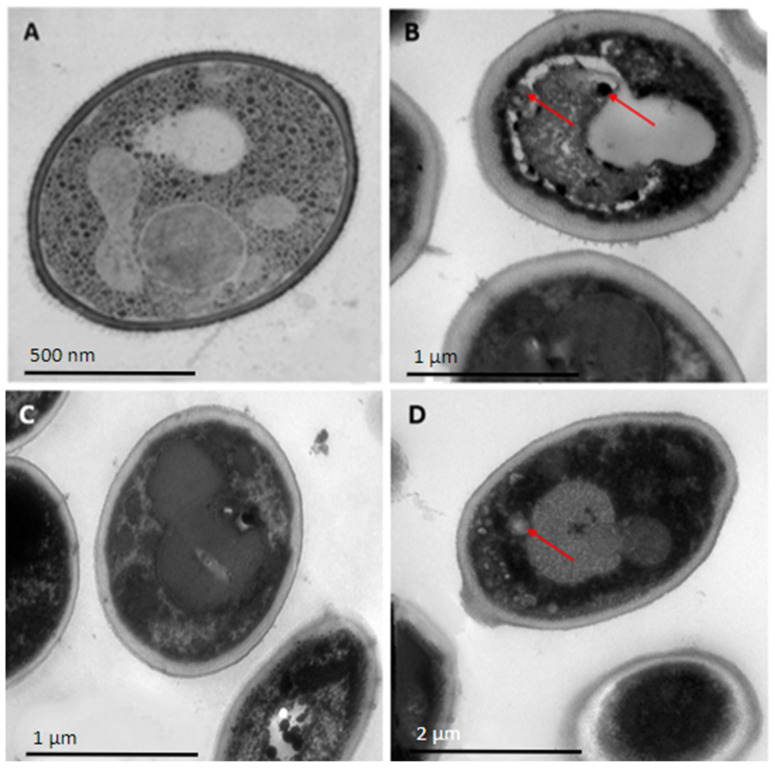
Ultrastructure of yeasts cells of *Candida auris* H0059-13-1421, untreated (**A**) and treated, with 100 µM of each of the peptides PNR20 (**B**), PNR20-1, (**C**) and 35409 (**D**). Red arrows show morphological alterations. Microphotographs were taken with Gatan camera in a transmission electron microscope (TEM).

**Figure 3 antibiotics-12-01234-f003:**
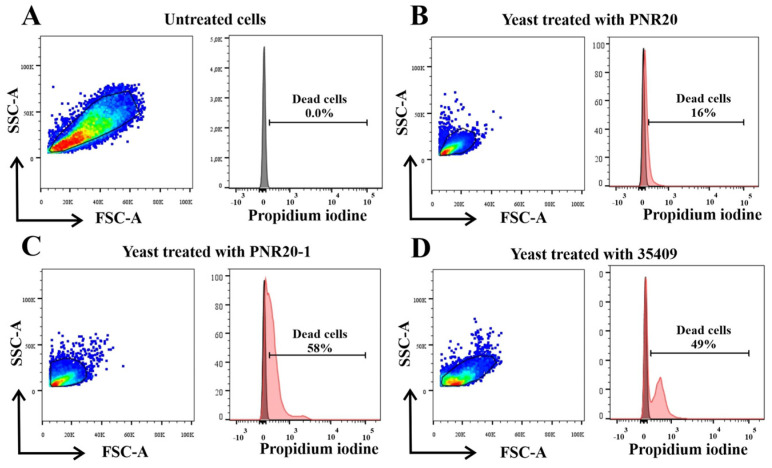
Cell viability of *Candida albicans* ATCC 10231, untreated cells (**A**) and after treatment, with 100 µM of each of the peptides PNR20 (**B**), PNR20-1 (**C**), and 35409 (**D**).

**Figure 4 antibiotics-12-01234-f004:**
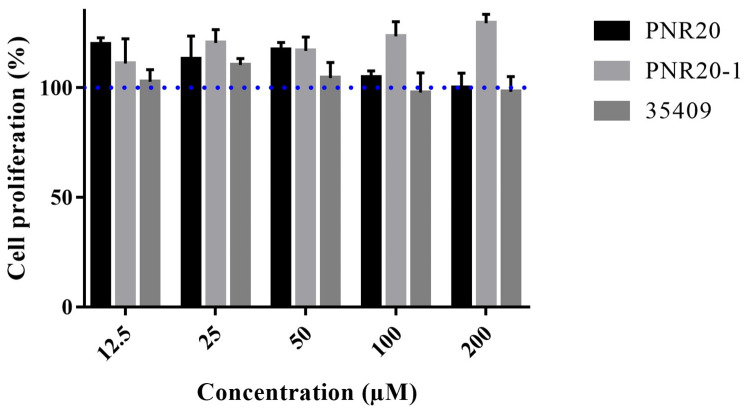
Cytotoxicity of 200 µM of each of the peptides PNR20, PNR20-1, and 35409 on L929 murine fibroblasts. A hundred percent (100%) growth is indicated by the blue dotted line.

**Table 1 antibiotics-12-01234-t001:** Antifungal activity of peptides PNR20, PNR20-1, and 35409 against *Candida* species.

Strain	Peptide	MIC	IC_90_	IC_50_
*C. albicans* ATCC 10231	PNR20	100 µM	50 µM	6.25 µM
	PNR20-1	100 µM	ND	6.25 µM
	35409	>100 µM	>100 µM	25 µM
*C. parapsilosis* ATCC 22019	PNR20	100 µM	25 µM	6.25 µM
	PNR20-1	>100 µM	50 µM	6.25 µM
	35409	>100 µM	100 µM	1.56 µM
*C. glabrata* ATCC 2001	PNR20	100 µM	ND	6.25 µM
	PNR20-1	>100 µM	>100 µM	3.12 µM
	35409	>100 µM	>100 µM	100 µM
*C. krusei* ATCC 6558	PNR20	25 µM	12.5 µM	3.12 µM
	PNR20-1	25 µM	ND	ND
	35409	100 µM	50 µM	3.12 µM
*C. tropicalis* ATCC750	PNR20	50 µM	25 µM	6.25 µM
	PNR20-1	100 µM	50 µM	25 µM
	35409	50 µM	ND	ND
*C. auris* H0059-13-1421 FCZ susceptible	PNR20	>100 µM	>100 µM	25 µM
	PNR20-1	>100 µM	>100 µM	6.25 µM
	35409	>100 µM	>100 µM	12.5 µM
*C. auris* H0059-13-2220 FCZ and AMB resistant	PNR20	>100 µM	>100 µM	25 µM
	PNR20-1	>100 µM	>100 µM	100 µM
	35409	>100 µM	>100 µM	>100 µM
*C. auris* H0059-13-2251 FCZ and AMB resistant	PNR20	>100 µM	>100 µM	100 µM
	PNR20-1	>100 µM	>100 µM	>100 µM
	35409	>100 µM	>100 µM	50 µM
*C. auris* H0059-13-2276 FCZ and AMB resistant	PNR20	>100 µM	>100 µM	100 µM
	PNR20-1	>100 µM	>100 µM	100 µM
	35409	>100 µM	>100 µM	>100 µM
*C. auris* H0059-13-2265 FCZ and AMB resistant	PNR20	>100 µM	>100 µM	100 µM
	PNR20-1	>100 µM	>100 µM	>100 µM
	35409	>100 µM	>100 µM	>100 µM

MIC: minimum inhibitory concentration. IC_90_: inhibitory concentration 90%. IC_50_: inhibitory concentration 50%. FCZ: fluconazole; AMB: amphotericin B. ND: not determined.

**Table 2 antibiotics-12-01234-t002:** Antifungal activity of peptides PNR20, PNR20-1, and 35409 against biofilm formation in *Candida albicans* and *Candida auris*.

Time	Species	Peptide	BMIC	IC_90_	IC_50_
24 h	*C. albicans* ATCC 10231	PNR20	12.5 µM	ND	ND
		PNR20-1	25 µM	12.5 µM	3.12 µM
		35409	≥100 µM	≥100 µM	3.12 µM
	*C. auris* H0059-13-1421	PNR20	ND	ND	25 µM
		PNR20-1	ND	ND	25 µM
		35409	ND	ND	12.5 µM
72 h	*C. albicans* ATCC 10231	PNR20	25 µM	ND	ND
		PNR20-1	100 µM	ND	50 µM
		35409	ND	ND	50 µM
	*C. auris* H0059-13-1421	PNR20	ND	ND	25 µM
		PNR20-1	ND	ND	12.5 µM
		35409	ND	ND	25 µM

BMIC: biofilm minimum inhibitory concentration. IC_90_: inhibitory concentration 90%. IC_50_: inhibitory concentration 50%. ND: not determined.

## Data Availability

The data presented in this study are available on request from the corresponding author.

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
