# Peer review of "In Vitro Antifungal Activity of Three Synthetic Peptides against Candida auris and Other Candida Species of Medical Importance"

_antibiotics, 2023, doi:10.3390/antibiotics12081234_

Round 1
Reviewer 1 Report
I read the manuscript with great interest since resistance to antifungals is a worrisome reality that threatens the effective control of systemic infections caused by fungi, especially in hospital settings.
I found the manuscript to be well written, concisely and convincingly demonstrating the in vitro antifungal activity of three synthetic antimicrobial peptides (PNR20, PNR20-1 and 35409), that have previously been shown to have also antibacterial effects, against Candida species.
In general, I consider that the conclusions expressed by the authors are consistent with the presented results, where they demonstrate an evident activity, particularly of the PNR20 peptide, against C. albicans, C. parapsilosis, C. glabrata, C. krusei and C. tropicalis; in addition to moderate activity against various C.auris isolates (fluconazole susceptible and fluconazole-amphotericin B resistant). Authors also present a concise introduction with relevant information, providing the readers enough background, as well as a compelling discussion of the results.
I have only a few minor observations:
-Figure 3 and Figure 4 are mislisted.
-The results presented in section 2.4, about the percentage of dead cells, do not match with those presented in the corresponding figure, or if those % are the mean of several experiments, please explicit it.
As a suggestion, besides extending your research to clinical isolates as stated in the discussion, it would be interesting to explore possible synergic interactions between the AMPs or in combination with the antifungals, specially in those strains that showed less susceptibility to one AMP alone.
Author Response
Dear reviewer, we appreciate each of your comments. We will answer your concerns below:
- Thanks for the remark, the figures numbers have been corrected.
- This information was corrected in the manuscript
- We agree with the reviewer's comments. Currently, we are further exploring the antifungal activity of these antimicrobial peptides in association with antifungals such as fluconazole. This information was added in the last paragraph of the discussion.
Reviewer 2 Report
The article, although not groundbreaking, contributes some important information about the antimicrobial activity of the tested peptides. The introduction is a bit too long, Information on the epidemiology of C. auris and other candidiasis is not necessary. Reference 17 refers to an article sent to the editor and, as I understand it, still under review. This raises questions.
Author Response
Dear reviewer, we appreciate each of your comments. We will answer your concerns below:
- We appreciate the feedback. However, we consider this information important since it emphasizes the importance of infections caused by Candida species. Actually, reviewer #3 suggested including additional data in this section to complement the information on the epidemiology of these infections.
- We agree and understand the reviewer's concern regarding the sequences of the PAMs included in this study. However, we consider that it is sensitive information since the molecule sequences are in the patent process. For this reason, we decided to remove reference 17 and write unpublished results instead. We would like to clarify that 3 authors of reference 17, which is already in publishing process, are part of this manuscript too.
Reviewer 3 Report
1. Including the global burden or morbidity and mortality caused by Candida auris is suggested
2. In Figures 1 to 4, what (100 uM ??) were the concentrations of AMP (PNR20, ... ) used for treating cells or other tests?
3. Figure 3: What was the control (positive and negative) for the cell proliferation assay? Cell proliferation exceeds 100% at lower concentration of peptides, does this means these peptides are growth promoter or enhancer?
Some minor typographical error such IC50 (IC<superscript>50; IC50), line 441 (1.5 mL)
Author Response
Dear reviewer, we appreciate each of your comments. We will answer your concerns below:
- We appreciate the reviewer´s comments and included the requested information in the introduction section (reference 3).
- Thanks for your suggestion. Concentrations used in each experiment have been included in the corresponding figure legend.
- We appreciate the reviewer's comment. In the cell proliferation assay we used L929 murine fibroblasts. In the case of the growth control group, the cells did not receive any treatment or stimulation with PAMs (blue line). In response to the reviewer's concern, we consider that these peptides are not growth enhancers because there is no significant difference in the proliferation of cells treated with the peptides compared to this control.
-
Thanks for the remark, this was corrected in the manuscript.